

# In my remembered country: what poetry tells us about the changing perceptions of volcanoes

*Arianna Soldati[1] & Sam Illingworth[2]*

*[1]Department of Earth and Environmental Sciences, Section for Mineralogy, Petrology and Geochemistry, Ludwig Maximilian University of Munich, Munich, Germany*

*[2]Department of Natural Sciences, Manchester Metropolitan University, Manchester, UK*

**Correspondence:** arianna_soldati@hotmail.it





# Abstract

In this study we investigate what poetry written about volcanoes from 1800 to the present day reveals about the relationship between humanity and volcanoes, including how it evolved over that time frame. In order to address this research question, we conducted a qualitative content analysis of a selection of 34 English-language poems written about the human-volcano interactions. Firstly, we identified the overall connotation of each poem. Then, we recognized specific emerging themes and grouped them in categories. Additionally, we performed a quantitative analysis of the frequency with which each category occurs throughout the decades of the dataset. This analysis reveals that a spiritual element is often present in poetry about volcanoes, transcending both the creative and destructive power that they exert. Furthermore, the human-volcano relationship is especially centred around the sense of identity that volcanoes provide to humans, which may follow from both positive and negative events. These results highlight the suitability of poetry as a means to explore the human perception of geologic phenomena. Additionally, our findings may be relevant to the definition of culturally appropriate communication strategies with communities living nearby active volcanoes.

**Graphical Abstract**

Not a god,
But unaffected
As we swarm like ants
Across your convex canopy;
Emerging from our shared landscape
To create new identities that stretch beyond the
Rich and fertile soils that bask beneath your presence.
Your destructive nature respected and then considered, as we
Celebrate the otherness of your existence; naïve in our assumptions
That you would sense our feats beneath the shadows of your magnificence.



# 1. Introduction

We live on a geologically active planet, and volcanoes are one of the most visually impressive geological features of our world. Whereas most geologic phenomena (e.g. orogeny, erosion) occur slowly over a timescale of millions of years, being *de facto* invisible to us humans, volcanic activity is a fast process that can radically change the natural landscape within a matter of hours, directly affecting the nearby communities. There are 150 active volcanoes in the world, and at any given time, about 20 of them are erupting (Durant et al., 2010;Venzke, 2013). The range of volcanic activity is broad (effusive to explosive), and so is the spectrum of feelings they inspire (from awe to fear). Certainly, they do not leave us indifferent.

The areal impact footprint of eruptions depends on their Volcano Explosivity Index (VEI). In past times, only large eruptions affecting people (i.e. occurring in inhabited areas) would have been reported and become known worldwide. Large eruptions (VEI>4) statistically occur every few years, and have footprints of tens of kilometres (Brown et al., 2015). At the beginning of the 19th century there were 10 million people worldwide living within 100 km of volcanoes that have erupted during the Holocene (Siebert et al., 2011); nowadays, that number has increased to 800 million.

Volcanoes constitute a fundamental part of the Earth system, recycling elements from the mantle to the surface and the atmosphere. But cohabiting with them comes with a unique set of challenges and rewards for local communities. Understanding how people perceive volcanoes is fundamental in shaping effective scientific communication strategies, including in times of crises (see e.g. Nave et al., 2016;Donovan et al., 2018;Avvisati et al., 2019). These perceptions may in turn be affected by a variety of factors, including: scientific knowledge, spiritual beliefs, oral history, and personal experience (direct or indirect), which vary widely in space and time across different cultures and historical periods.

Monitoring people's perception of volcanoes is difficult. Psychologists sometimes work with local communities in the aftermath of eruptions (Paton et al., 2000;Sword-Daniels et al., 2018;Paton, 2019), yet while these works have great merit, they mainly focus on a very small, specific part of the human-volcano interaction (i.e. the instances in which volcanic activity clashes with human life), and therefore paint a partial, skewed view of this relationship. Additionally, these interventions have only become common practice in recent times, and therefore cannot shed light on the evolution of the relationship between humans and volcanoes over time

A potential medium through which a more nuanced and expansive dataset of this relationship can be established is the arts. The arts might be thought of as ubiquitous expressions of human nature, and have often been inspired by volcanoes, with notable examples including paintings by Turner (Daly, 2011), Warhol (Sigurdsson,



2015), and Munch (Olson et al., 2007). We can conceivably use art pieces to guide us in our understanding of how volcano-human relations have changed through time. For example, previous research has shown how the colours of sunsets painted by famous artists can be used to estimate vog (i.e. smog or haze containing volcanic dust and gases) concentrations in the Earth's past atmosphere, revealing that ash and gas released during major volcanic eruptions enhanced sunlight scattering, making sunsets appear redder, as reflected by the paintings that were produced in the aftermaths of such events (Zerefos et al., 2014). Similarly, studies have been conducted which have: dated volcanic eruptions through Neolithic cave paintings (Schmitt et al., 2014), investigated the extent to which artists have used volcanoes to represent societal upheaval and transformation (Sivard, 2011), and even demonstrated how drawing volcanoes can be used to empower children in informal learning environments (Weier, 2004). These examples concern mainly drawings and paintings, aside from which there has been a relative paucity in research that investigates how the arts might be used to shed light on changing perceptions of volcanoes, and how this information can be directly useful in the communication of volcanic hazards and risk in the modern era..

In this study, we begin to address this gap in the literature by choosing to focus on another form of art: poetry. While attempts to define what is and what is not poetry can be contentious (see e.g. Ribeiro, 2007), here we take poetry to be any written form that is composed by the line. We consider prose to be differentiated from poetry, in that prose is instead composed by sentences and is written with standard grammatical structure. This line-by-line composition of poetry means that it is able to convey meaning in a concise manner, which in turn lends itself to analysis and interpretation. Poetry provides a powerful medium through which to interpret human behaviours and perceptions, and an analysis of poetry as data has been used to provide understanding on topics ranging from living with HIV (Poindexter, 2002) and dementia (Zeilig, 2014), to attitudes relating to environmental change (Illingworth and Jack, 2018) and compassion fatigue in nurses (Jack and Illingworth, 2017). In this study, we aim to build on this research, using poetry as a medium through which to investigate perceptions of human-volcano interactions.

By conducting a detailed qualitative content analysis for a selection of volcano poetry, this study aims to understand how poets have interpreted the relationship between humans and volcanoes from the beginning of the nineteenth century to the modern day. In Section 2, we discuss the methodology that was utilised in this study, outlining why this approach was adopted and establishing its validity and reliability. Section 3 contains a discussion of how categories and themes emerged from our analysis, and how this relates to the research question defined in Section 2. Finally, Section 4 contains the conclusions of this study, along with future directions for further research.



# 2. Methodology

The methodology that we have adopted in this study involved treating poetry as data, allowing for text to be analysed with respect to attitudes relating to volcanoes. While several other methods exist for the analysis of textual data (e.g. ethnography, phenomenology, grounded theory, etc.), we have chosen qualitative content analysis because of its ability to highlight both the context and the content of the chosen text, which for a subjective medium such as poetry is essential.

The methodology that is adopted in this study largely follows that of Illingworth (in preparation), insofar as the qualitative content analysis of the poetry is guided by these six steps: formulation of research questions; selection of poetry to be analysed; definition of codes and categories to be applied; outline and implementation of coding process; determination of trustworthiness; and analysis of results. The first five of these steps are outlined below, with the analysis of the results presented in Section 3.

We note here that the poems that were analysed in this study are presented according to the following format: 'Poem Name' by Author Name (Year of Publication), and that links to the full texts of all these poems can be found in Soldati and Illingworth (2019).

## 2.1 Formulation of research question

By performing a qualitative content analysis on poetry that has been written about volcanoes, but critically not for the purpose of research, this study aims to better understand the way in which poets have interpreted the relationship between humans and volcanoes. For the purposes of this study, this has been formulated into the following Research Question (RQ):

RQ: what does poetry written about volcanoes reveal about the relationship between humanity and volcanoes?

## 2.2 Selection of poetry to be analysed

We began by selecting poetry that mentioned or featured volcanoes in some capacity. In doing so we set ourselves three constraints. Firstly, the poetry must be written in English, or else have an author-sanctioned English translation; this constraint was because English is the only common language that is shared by the authors of this study. Secondly, the poems must have been written from 1800 onwards; this constraint was introduced to make the list more manageable for analysis purposes, and because this was when many consider volcanology to have first been fully established as a scientific discipline through the research of Ben Franklin, James Hutton, Alexander von Humboldt, and others (Sigurdsson, 2000).



Finally, where possible, we aimed to find at least one poem per decade, in order to
try and better capture any change in attitudes from the early nineteenth century to
the present day. Given these constraints, we performed a manual Internet search for
poetry that mentioned the word 'volcano' or 'volcanoes' and which, in the first
instance, came from recognised poets and/or well-established poetry
journals. Additionally, we reached out to the volcanology research community
through the volcanology listserv (volcano@lists.asu.edu) asking for further
suggestions.
At this stage, we had a list of 53 poems, spanning every decade from the 1800s to
the 2010s. The two authors then independently read these poems and filtered them
according to two further criteria: only poems that were 100 lines or less were to be
considered, and any poet could only have one of their poems considered for further
analysis. The line limit was introduced in order to make the analysis more
manageable, and a limit of one poem per poet was introduced to allow for a wider
selection of voices to be considered, and also to improve the validity of the approach
in terms of the triangulation of data (see Section 2.5). In instances where there were
multiple poems from a single poet, we agreed upon the one that we both found to be
the most aesthetically pleasing, prior to any formalised content analysis. Poems that
were removed because of length included 'The last days of Herculaneum' by Edwin
Atherstone (which is over 500 lines long), while poets with more than one poem
included Emily Dickinson (six poems, for whom 'I have never seen "Volcanoes"' was
chosen) and PB Shelley (two poems, for whom 'The Cloud' was chosen). Following
the application of this further selection criteria, 41 poems remained available for
analysis.
At this stage, both authors independently read each of the poems and categorised
them as being either 'Positive', 'Negative', 'Positive & Negative', 'Neutral', or 'Invalid'
in their portrayal of volcanoes from a human perspective (in line with RQ), the results
of which can be seen in Table 1.
Performing this analysis enabled each of the authors to better familiarise themselves
with the poetry, and also highlighted any further poems that needed to be removed
from the study, because they were 'Invalid', i.e. those poems that did not concern the
relationship between volcanoes and humanity. Following this individual analysis (the
rows corresponding to 'AS' and 'SI' in Table 1), the two authors met up to exchange
their analyses and agree upon a broad classification of the poems in terms of
connotation ('Both' in Table 1). This broad categorisation was useful for three
reasons. Firstly, it revealed several differences in how the poems had been
individually categorised, highlighting how these differences could be discussed and
developed, thereby helping to improve the trustworthiness of the subsequent in-
depth content analysis by ensuring investigator triangulation (see Section 2.5).
Secondly, it revealed that this broad categorisation of the connotations of the poems
as either: 'Positive', 'Negative', 'Positive & Negative', or 'Neutral' was too broad to





offer any in-depth analysis with respect to the RQ. Finally, this approach revealed
additional poems that should be removed from the study because they did not
obviously concern the relationship between volcanoes and humanity. For example,
'Inside the Ghost Volcano' by Will Alexander (1998) was removed at this stage
because the volcano was being used as an abstract metaphor from which no
perspective of the perceived relationship between volcanoes and humanity could be
directly observed. In total a further 7 poems (the 'Invalid' column in Table 1) were
removed following this step, resulting in a total of 34 poems on which to perform a
more in-depth content analysis. This further selection criteria meant that two
decades were now absent from our study (1940 and 1960). However, given that
every other decade from the 1800s onwards was still present we were satisfied that
we had a sufficient temporal representation of poems to conduct a detailed content
analysis with respect to the RQ.
**2.3 Definition of codes and categories to be applied**
A conventional approach to qualitative content analysis was adopted in this study,
with preconceived categories being avoided, and categories being determined by the
implementation of the coding process instead (see Section 2.4). While in some
instances a directed content analysis might be more appropriate, this methodology is
usually used in those instances where an existing theory would benefit from further
description (Hsieh and Shannon, 2005). As the research question to be addressed in
this study is unique, a directed approach is inappropriate. Similarly, a summative
content analysis (i.e. one in which the frequency of words and/or phrases was only
quantitatively assessed) would fail to fully account for the context of the poetry
alongside its content.
**2.4 Outline and implementation of coding process**
Both authors individually read each of the 34 poems, and assigned codes to sections
of the text (Table 2). As each new code emerged we individually went back through
the poems that had previously been coded to check whether these also contained
any lines that could be labelled with any newly emergent code. After coding all of the
poetry in this manner, we independently read all of the poems in full again and made
sure that each of them had been coded accurately and that a saturation of emergent
codes had been reached, thereby improving the trustworthiness of the approach
(see Section 2.5). As can be seen from Table 2, this resulted in a total of 28 codes (9
for SI and 19 for AS). We then read each of the poems again to make sure that no
coding had been missed.
Following this independent coding process, the two individual codebooks (one for AS
and one for SI) shown in Table 2 were combined in order to search for emergent
categories. This categorisation of the individual codes was initially done by SI, before



being modified by AS, and then ratified by both SI and AS, with both authors
agreeing upon the five emergent categories (and corresponding codes) shown in
Table 3: 'Landscape', 'Identity', 'Destruction', 'Spiritual', and 'Creation', each of which
will be discussed in detail in Section 3. After these codes had been grouped as such
we went back through each of the individual occurrences (e.g. the 61 segments of
poetry that were categorised as 'Landscape') to make sure that they did indeed
belong in this category.
In this combination of codebooks to search for emergent categories, we found that
some of our early individual coding had been done erroneously, e.g. in 'Ice Child' by
John Haines (1999), the following section was coded as 'Identity' by SI and
'Destruction' by AS; however, in the merger of our codebooks we decided that a
more appropriate category was actually 'Landscape', which as can be seen from
Table 3 was not associated with either of these codes:
we find your interrupted life,
placed here among the trilobites
and shells, so late unearthed.
Following this categorisation, each of the five categories shown in Table 3 were
further examined for any theme(s) that expressed underlying meaning in relation to
RQ, the results of which are presented in Section 3.7.

## 2.5 Determination of Trustworthiness

At each stage of the qualitative content analysis that was adopted in this study, the
individual codes and categories were re-examined in order to confirm that they
accurately captured the poetry in relation to RQ. Each author carried out this coding
independently, until there were no further codes or categories found to be emerging
from the data, i.e. until descriptive saturation had been reached (Lambert and
Lambert, 2012). Triangulation as a validation strategy (Flick, 2004) was achieved by
using data drawn from different times and places (the poems) and conducting an
analysis using two different investigators (the authors). This use of systematic
sampling, triangulation and constant comparison, and proper audit and
documentation (see Section 2.2 and 2.4) combined to ensure both the reliability (i.e.
the consistency with which this analysis would produce the same results if repeated)
and the validity (i.e. the accuracy or correctness of the findings) of this approach
(Leung, 2015). Given that the analysis of the poetry as described here represents a
somewhat subjective approach, the reliability of the findings might be called into
question. However, as outlined by Morse et al. (2002), our methodological
coherence, sampling strategy, and saturation of emergent codes ensures the
reliability, and thus repeatability, of our approach in this qualitative analysis.



## 3. Results and Discussion

As can be seen from Table 3, five major categories emerged from the thematic
coding of the poems used in this study. We will now discuss each of these emergent
categories, how they relate to RQ ('what does poetry written about volcanoes reveal
about the relationship between humanity and volcanoes?'), and how these findings
compare to other research that has been conducted into the changing perceptions of
volcanoes (Section 3.1 - Section 3.5).
Following a discussion of these categories we present a quantitative analysis of the
poems in terms of how the frequency of these emergent categories have changed
over time (Section 3.5), followed by a presentation of the overall theme that emerged
from conducting this analysis, and how this relates to RQ (Section 3.6).

### 3.1 Landscape

Landscape is the most recurrent of the five emergent categories, being present in
every poem that was analysed, regardless of when it was written. Here volcanoes
serve primarily as a backdrop; they are part of the environment where the poem is
set, typically a prominent feature that dominates the landscape. For example, in
'Hunched Back Volcano' by Genevieve Taggard (1914), the poet paints a particularly
vivid image of an active volcano, anthropomorphised as having a passionate
(erupting) mouth (crater), set against a backdrop of stars:
Red is the mouth of Pele, passionate
Against the fires of the kindling stars:
Fire to fire moves: the heavens wait
By contrast, in 'Mount Broadshield' by Jónas Hallgrímsson (1841) the volcano is now
quiet and peaceful - an image highlighted by the presence of snow - though it still
dominates the landscape with its (regal) presence:
Queen of all our country's mountains,
crowned with snow sublime and pure!
Once you poured from fiery fountains
floods of lava down the moor.

In 'Flowers from the Volcano' by Claribel Alegría (2013), the volcanic crater,
presumably at least temporarily inactive, becomes part of the human landscape,
representing the place where the flower growers of the poem now live:

Farther up, in the crater



within the crater's walls
live peasant families
who cultivate flowers
their children can sell.
Even in its absence, the volcano can still dominate the landscape. For example, in
'Road Trip' by Vijay Seshadri  (2016), it is not the volcano itself which provides the
poem's setting, but rather the beaches that came from its erosion, with their black
sand and tide pools excavated in its obsidian lava:
Tomorrow or the day after or the day after that,
on the volcano beaches fringed with black sand
and heaped with tangled beds of kelp,
by the obsidian tide pools that cradle the ribbed limpet
and the rockbound star,
These poems also capture the effect that volcanoes have in changing the landscape,
such as in 'Peace' by D H Lawrence (1929), which provides a description of an
effusive eruption; firstly, as it occurs:
Brilliant, intolerable lava
Brilliant as a powerful burning-glass
Walking like a royal snake down the mountain to-
wards the sea.
And then, after the eruption has ceased:
Peace congealed in black lava on the doorstep.
Within, white-hot lava, never at peace
Till it burst forth blinding, withering the earth;
To set again into rock
Grey-black rock.
These descriptions of the changing landscape also highlight how the language that is
used by poets in their descriptions can both reinforce and counteract scientific
terminology and understanding. The likening of lava to glass is scientifically
accurate, as both lava and glass are examples of a silicate melt. In contrast to this,
the use of colour in this passage, whilst evocative, is not actually grounded in reality:
lava does indeed turn into black or grey-black rock as it cools down, but is never
white-hot. The colour choice may have been dictated by a contrast (black vs. white)
or by the idea, pushed further than it actually occurs in reality, that the hotter
something gets the closer it gets to white (with lava typically stopping at red-orange),
and the idea of hot, molten lava still persisting within a cooled, solid outer crust does
in fact accurately reflect the inward cooling process of lava flows. This is of course



not to say that poems should ensure scientific accuracy at the cost of their
aesthetics; we draw attention to these scientific inaccuracies only to highlight that we
are not interested in how accurately these poems capture volcanic behaviour, but
rather what the poetry can tell us about the relationship between humanity and
volcanoes. This use of poetry, mirrors that which Aristotle observed in the *Poetics,*
i.e. that whilst history deals with specific events, poetry deals with general truths
(Yanal, 1982).

The fact that volcanoes feature so prominently as a setting in these poems highlights
that the poets perceived very clearly their relationship with humanity. Similarly, the
language used in the poetry acts to position volcanoes as an awesome part of our
shared landscape, perhaps explaining in part why humans were first drawn to them
before they became valued for more tangible goods such as precious metals and
stones (Fisher et al., 1998), as well as soil fertility.

## 3.2 Identity

Volcanoes are often used to represent or highlight part of the poet's identity. They
are present in their childhood memories, as well as in family history or cultural
heritage. Identity is the second most recurrent category (after 'Landscape'), missing
from only two of the analysed decades: the 1860s and 1880s. This ubiquity is
extremely revealing of the profound tie between humans and volcanoes; here the
volcanoes are not merely passive elements of the landscape, but rather active
agents. Even when they are standing still, their presence is so impactful as to be
defining for the poet, as evidenced in 'Punctuation Marks' by Phillip Nanton (1992),
set on the author's native island of Saint Vincent (Saint Vincent and the Grenadines),
whose landscape is dominated by the active volcano La Soufrière:

> Come nearer, focus on one dot of an island
> I was born there, on the rim of a volcano
> on the edge of a large full stop
> where the sand is black
> where the hills turn a gun-barrel blue
> where the sea perpetually dashes at the shoreline
> trying to reclaim it all.

Similarly, in 'Flowers from the Volcano' by Claribel Alegría (2013), the poet's home
country of El Salvador is remembered nostalgically, its personal resonance defined
by the situation of its volcanoes:

> Fourteen volcanos rise
> in my remembered country
> in my mythical country.



In this poem, the volcanoes represent not only the literal volcanoes in that region, but
also the eruption of violence brought about by the civil wars of the 1970s and 1980s
across El Salvador, Nicaragua, and Guatemala, and how they too came to form a
part of the poet's life and identity.
This personalisation of the volcano is continued in 'Sonnet 5' by Pablo Neruda
(1959), for whom volcanoes become the distinctive feature of his native Chile,
elevated to the same rank as the love / lover which the poet found there:
10       and on through the streets like a man wounded,
11       until I understood, Love: I had found
12       my place, a land of kisses and volcanoes.
In many of these instances of identity, the volcano takes centre stage in a formative
childhood memory, such as in 'What For' by Garrett Hongo (1982), where the poet
expresses the extent to which his childhood identity was itself defined by the
presence of volcanoes:
I lived for the red volcano dirt
staining my toes, the salt residue
of surf and sea wind in my hair,
the arc of a flat stone skipping
in the hollow trough of a wave.
In contrast to this, 'Axis' by Ray Gonzalez (2015) captures the way in which a
particular volcano is a matter of family history, a mythical entity that the poet himself
did not experience personally, but which is nonetheless present in the defining
stories of his heritage:
30       The volcano in my grandmother's Mexican village
31       smothered the town, though the girl escaped because
32       the axis of revolution sent her family into exile,
Here this identity is not necessarily a positive one: the forced exile of the poet's
grandmother is remembered as an environmental revolution which they were
powerless to avoid, but which became a defining aspect of both the grandmother
and her grandson's assumed identity.
Identity is a nuanced concept, and has both positive and negative connotations
associated with it. However, what is evident from these poems is that these
associations are typically not purely adverse, and even when they are they form
important societal ties that have come to define both past and current generations.
Although the destructive impacts of volcanoes are well documented and recognized,
many of the possible societal and cultural benefits are not always fully considered



(Kelman and Mather, 2008), and the emergence of identity as a key category across the majority of the analysed poems serves to further highlight this disparity.

## 3.3 Destruction

This category considers those poems that make reference in some part to the destructive power of volcanoes, in relation to humankind. Poems that were categorised as such include those that make reference to physical damage of societal assets or destruction to human beings, such as is evident from this extract from 'Blankets of Blood' by Arthur Rimbaud (1872):

> Oh my friends! —My heart knows its own brothers!
> Dark strangers, what if we were to leave? So leave! Leave!
> O misfortune! How the earth melts upon us,
> How I shake as it melts on me and you,

These poems also consider those that make reference to physical damage or destruction to property or land that is owned and/or cherished by humans, such as in this passage from 'Peace' by DH Lawrence (1929):

> Forests, cities, bridges
> Gone again in the bright trail of lava.
> Naxos thousands of feet below the olive-roots,
> And now the olive leaves thousands of feet below the
> lava fire.

This category also includes those poems that highlight the mental anguish of people caused by the negative effects of volcanoes, either directly through their eruptions or indirectly through the oppressive nature of living within close proximity to them, as can be seen in this extract from 'Negotiations with a Volcano' by Naomi Shihab Nye (1995):

> We need dreams the shape of lakes,
> with mornings in them thick as fish.
> Shade us while we cast and hook—
> but nothing else, nothing else.

The permeating sense of fear that is evident from this extract acts as a powerful testament to the severe mental anguish that experiencing a volcanic eruption and/or living near an active volcano can have on local residents, capturing these negative mental health effects in a manner that does not diminish them in comparison to more tangible examples of volcanic destruction. Research into the mental health impacts following volcanic eruptions have shown that nearby residents are at an increased risk of experiencing symptoms of post-traumatic stress disorder (Gissurardóttir et al.,



2019) and mental stress due to the loss (or threat of loss) of property (Horwell et al.,
2015). Indeed, such research has also highlighted the importance of implementing
psychosocial interventions with volunteers, both during and after a volcanic eruption,
in order to alleviate severe mental health risks (Espinoza et al., 2019). By
highlighting the mental distress that volcanoes can cause to humans, these poems
serve to highlight why this should not be neglected when considering the potential
impacts of volcanic activity.
Finally, this category also considers those poems in which volcanoes are responsible
for the fatality of human beings, as can be seen in these lines from 'Flowers from the
Volcano' by Claribel Alegría (2013):
Eternal Chacmol collects blood,
the gray orphans
the volcano spitting bright lava
and the dead guerrillero
and the thousand betrayed faces,
the children who are watching
so they can tell of it.
Here 'Chacmol' likely refers to a distinctive form of Mesoamerican sculpture
associated with sacrifice, and in this poem, the poet asks us to reflect on the role of
the volcanoes in shaping the landscape of Central America. Through death, several
of these poems invite us to consider how such sacrifices can result in new creation
(see Section 3.5), but they also serve as a reminder to the reader of the vast power
that these volcanoes possess, encouraging us humans to both respect and, at times,
fear their presence.
Given the severe destruction that volcanoes can wrought, as highlighted by this
emergent category, these poems might suggest that communities would not choose
to live in such environments of their own volition. However, when such destructive
incidents occur in the poetry, they rarely appear in isolation, and instead tend to
appear alongside other categories which represent potential benefits, not least
'Identity' (Section 3.2) and 'Creation' (Section 3.5). This contrast of categories
implies that in many instances the poets were intending to capture a mutuality of
both potential hazards and benefits. This interpretation further supports the findings
of other studies which have highlighted that many communities choose to expose
themselves to the negative consequences of volcanic hazards, so that they might
enjoy the benefits and opportunities that arise within the human–volcano system
(see e.g. Bachri et al., 2015;Stoffle et al., 2015).
**3.4 Spiritual**



As can be seen from Table 2 and Table 3, this category emerged from a single code,
which both authors independently labelled as such. This category represents those
poems that were considered to make reference to aspects of volcanoes that were of
spiritual significance. Some of the poems that were coded in this category make a
specific mention of religious figures, such as these references to Christ in 'Etna' by
Emily Pfeiffer (1889):
8       Martyr of mountains, shall I say, the Christ,
9       Bearing earth's sorrows, for its trespass made
Sin, that her sons may reap the fair increase
Of smiling fields? The offering hath sufficed:
And also, those in 'St Telemachus' by Alfred Tennyson (1892):
In the great name of Him who died for men,
Christ Jesus!' For one moment afterward
A silence follow'd as of death, and then
A hiss as from a wilderness of snakes,
Then one deep roar as of a breaking sea,
And then a shower of stones that stoned him dead,
And then once more a silence as of death.
Given the selection of English-language poems for this study, there is a strong
weighting towards poems that make reference to biblical figures from Christianity,
although other religious figures, such as the Buddha and several Mesoamerican
deities also feature, especially in those poems which were originally written in
another language, and yet for which an author-sanctioned translation exists (see
Section 2.2). For example, in 'A tale for Puuooo' by Taeko Jane Takahashi (2002),
we hear how:
Pele the goddess pierced and
thrusted, spilling lava into
Royal Gardens subdivision.
Here, Pele is the goddess of volcanoes and fire in the Hawaiian religion, where she
is also considered to be the creator of the Hawaiian Islands. In some instances, the
volcano itself is considered to be the chief religious figure or deity, such as in these
lines from 'Ice Child' by John Haines (1999), in which the volcano is explicitly
referred to as a now dormant god:
Under the weight of this mountain—
once a god, now only restless stone,
we find your interrupted life,
placed here among the trilobites



1    and shells, so late unearthed.

3    Or in 'Burning Island' by Gary Snider (1970), in which the volcano is given the status

4    of a spiritual creator and protector:

6        Volcano belly Keeper who lifted this island

7        for our own beaded bodies adornment

8        and sprinkles us all with his laugh—

At times this spirituality manifests itself with the poet (or their protagonist) appealing
to the volcano for either forgiveness or clemency, such as in 'Negotiations with a
Volcano' by Naomi Shihab Nye (1995), where the author pleads for mercy with the
volcano as they might do with another deity or spiritual figure:

15       Forgive any anger we feel toward the earth,

16       when the rains do not come, or they come too much,

17       and swallow our corn.

18       It is not easy to be this small and live in your shadow.

These lines also demonstrate how this associated spirituality serves to create an
'otherness', i.e. that volcanoes and humans might occupy the same landscape but
they are vastly different entities in terms of both temporal and spatial scales. The
separation that is imposed by this emergent spirituality also highlights potential
difficulties for cohabiting. There is at times a perceived unidirectional flow of power
that makes establishing a truly symbiotic relationship seem improbable, as can be
seen in the following lines from 'Ice Child':

Was it God—the sun-god of the Incas,

the imperial god of the Spaniards?

Or only the priests of that god,

self-elected—voice of the volcano

that speaks once in a hundred years.

The interpretations of volcanic eruptions have been shown to be interconnected with
the understanding of tradition by different religions and religious figures (see e.g.
Gaillard and Texier, 2010). The poems that feature in this category further
demonstrate that there is clearly a spiritual connection between volcanoes and
humans, whilst also serving to highlight the spiritual reverence in which volcanoes
are held, and the distance that this can introduce. By considering the spirituality of
volcanoes, these poems also ask us to contemplate the persistence of the spiritual
life system within a particular territory, aligning with previous research which has
shown how when humans push these boundaries in order to exploit them, the
system can become disrupted (Vitale, 2017).



**3.5 Creation**

The final category to emerge from this analysis is one of 'Creation'. Poems that feature in this category demonstrate how volcanoes can act as a creative (as opposed to, or in addition to being a destructive) force. For example, 'Darkness' by George Byron (1816) explores the aftermath and potential consequences of the 1815 eruption of Mt Tambora, Sumbawa, Indonesia, an event whose subsequent volcanic winter resulted in the infamous 'year without a summer' (Stommel and Stommel, 1979). Yet, despite the overall negative connotations of this poem, Byron still recognizes that prior to this eruption living alongside volcanoes was a positive experience, a source of happiness even:

> Happy were those who dwelt within the eye
> Of the volcanos, and their mountain-torch:

Another poem that demonstrates the creative potential of the volcano is 'Burning Island' by Gary Snider (1970), in which the poet asks us to consider the fertile soils that volcanoes provide, enabling a peaceful and profitable co-habitation:

> As we hoe the field
> let sweet potato grow.
> And as sit us all down when we may
> To consider the Dharma
> bring with a flower and a glimmer.
> Let us all sleep in peace together.

Many residents who live near volcanoes view eruptions as agents of change, often change for the good (Dove, 2008), which can in turn influence the extent to which they perceive the hazard and risk of living of living near to volcanoes. 'Burning Island' highlights the potential benefits of living in volcanic regions, where the rich volcanic soils can support highly productive agriculture (Rampengan et al., 2016). However, it also draws attention to the fact that the benefits of this relationship would appear to be unidirectional – volcanoes do not benefit from cohabiting the landscape with humans.

Continuing with notions of cohabitating, this category features those poems that demonstrate how such a relationship can be peaceful. For example, in 'Peace' by DH Lawrence (1929) the concept of placatory cohabitation is explored in the very first lines:

> PEACE is written on the doorstep
> In lava.



Here the potentially destructive lava gets all the way to the house, yet stops on the
doorstep, respectful of the human space. However, even here the potential dangers
of this cohabitation are clearly apparent: is this message from the volcano a genuine
offering of peace, or a warning shot across the bow; a reminder to humans of their
position in the hierarchical relationship that emerges through a consideration of the
'Spiritual' (Section 2.4)? 'Negotiations with a Volcano' by Naomi Shihab Nye (1995)
further explores the fragile nature of this cohabitation, explicitly wishing for the
volcano to remain dormant, and offering to be peaceful themselves if this wish is
granted:

12   We would be happy if you slept forever.

13   We will tend the slopes we plant, singing the songs

14   our grandfathers taught us before we inherited their fear.

15   We will try not to argue among ourselves.

In considering the potential for a creative and peaceful cohabitation of the landscape,
this category demonstrates how such a relationship is conditional, and is reliant on
the volcanoes to 'behave' in a certain manner towards (or with respect to) humanity.
**3.6 Quantitative Analysis**
After performing this qualitative content analysis, we quantitatively analysed how
both the connotation of volcanoes ('Neutral', 'Positive', 'Negative', 'Both') as well as
the frequency of the five emergent categories evolved through time in our dataset. In
order to do so, we aggregated our dataset by decade, and normalized the identified
connotations and categories to the number of poems in which they were represented
(Table 4). Furthermore, we considered the major eruptions which occurred in the
Northern Hemisphere (where the majority of the poets used in this study lived)
across the considered time frame (Table 5), and correlated these with the
connotations and categories of the dataset. The purpose of this quantitative analysis
was not to apply a statistical interpretation of the volcano-human interactions of the
analysed poetry; rather it was read alongside the qualitative analysis in support of
the emergent theme (Section 3.6) that arose from a consideration of these poems.

As can be seen from Fig. 1, poems in which volcanoes have either a 'Neutral' or
'Both' a positive and negative connotation, and therefore do not express a partial
value judgement, can be found consistently throughout the time span considered
(i.e. from the 1800s to the 2010s). The same can be also said for poems where
volcanoes are depicted with an overall 'Negative' connotation. In contrast, a
'Positive' representation of volcanoes only emerges in the 1900s, and does not
become frequent until the 1970s.





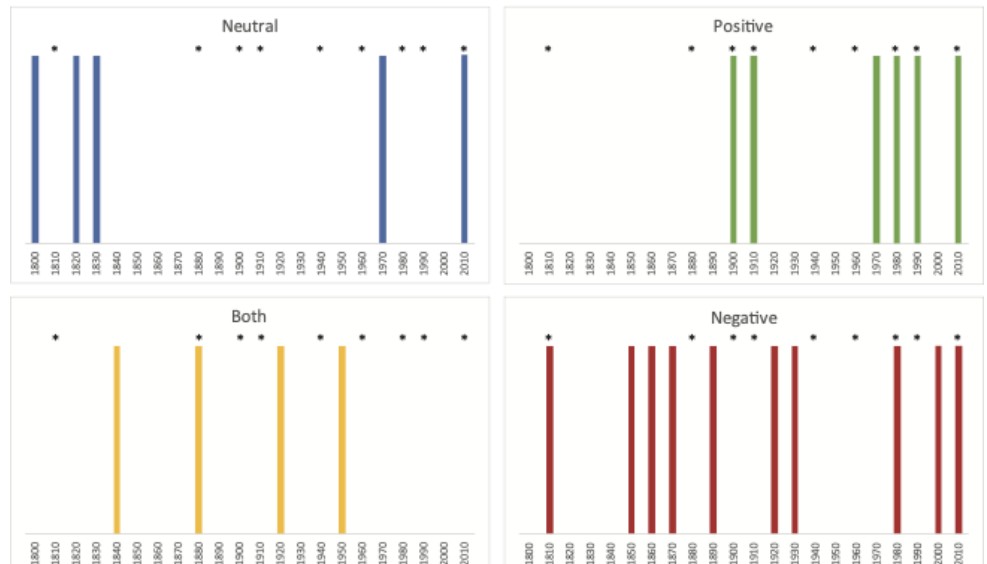

*Figure 1. Connotations ('Neutral', 'Positive', 'Negative', or 'Both') of the considered*
*poems, arranged chronologically and aggregated by decade. Asterisks indicate*
*major eruptions (as shown in Table 5).*
Plotting major eruptions alongside the associated connotations of the selected
poems (Fig. 1) does not reveal any obvious correlation. This can be attributed to the
fact that the vast majority of the analysed poems do not focus on specific volcanic
eruptions, but rather that they consider human-volcano interactions more generally.
One notable exception is the poem 'Darkness' by George Byron (1816), which as
discussed in Section 3.5 refers to the eruption (Volcanic Explosivity Index (VEI) of 7)
of Mount Tambora in 1815, and the subsequent 'year without summer'. During this
eruption, volcanic ash was injected high into the stratosphere, where it circled the
globe and persisted for a considerable time, visibly filtered the sunlight, and causing
a negative thermal anomaly of at least half a degree Celsius in the Northern
Hemisphere, as well as a famine that ultimately killed over 70,000 people (Brohan et
al., 2016). This event inspired artists in all fields (e.g. Turner's famous sunset
paintings, and Mary Shelley's *Frankenstein*), with Byron's poem also bearing witness
to this event:
I had a dream, which was not all a dream.
The bright sun was extinguish'd, and the stars
Did wander darkling in the eternal space,
Rayless, and pathless, and the icy earth
Swung blind and blackening in the moonless air;
Morn came and went—and came, and brought no day,



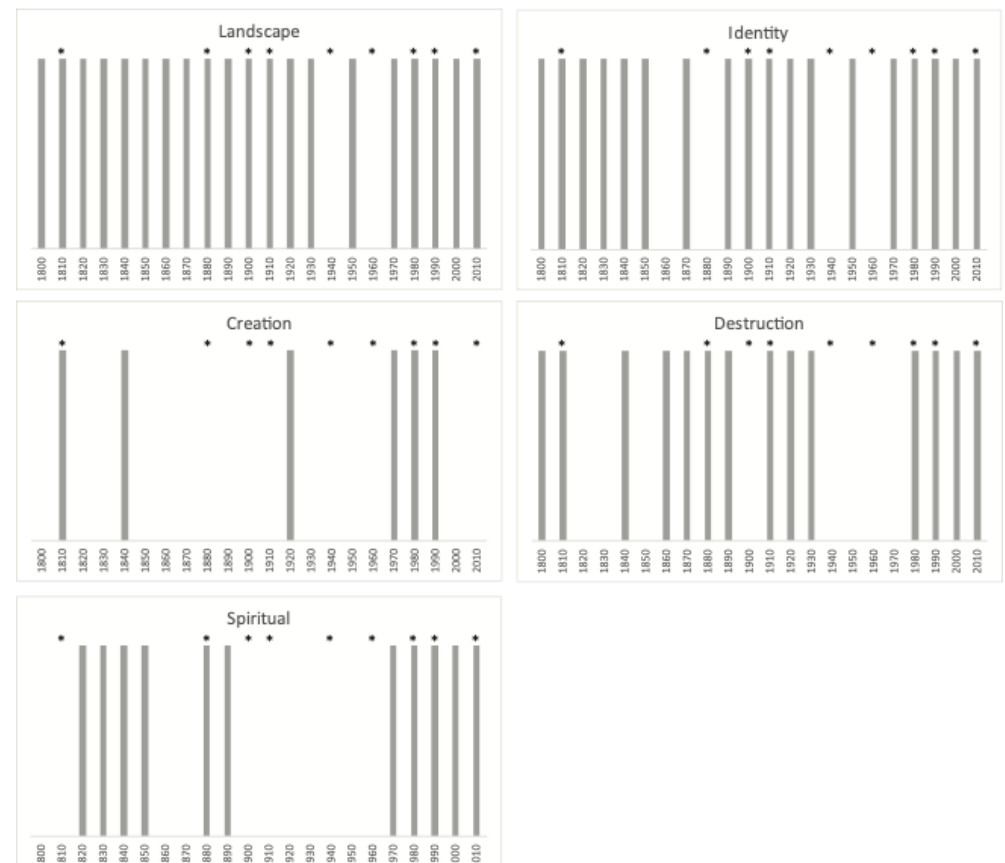

*Figure 2. Emerging categories in the considered poems, arranged chronologically*
*and aggregated by decade. Asterisks indicate major eruptions (as shown in Table 5).*
At the categories scale (Fig. 2), we see that the volcano-human interactions in
relation to the 'Landscape' occur in all represented decades (as discussed in Section
2.2 the 1940s and 1960s are not represented in this analysis). The notions of
'Identity' that are associated with these interactions are almost as ubiquitous, with
the only exceptions being the 1860s and 1880s.
As can be seen from Fig. 2, the 'Destruction' in the volcano-human interactions of
the selected poetry is represented more frequently across the considered time period
than the 'Creation' of such interactions. There is also somewhat of a correlation
between 'Destruction' and each of the major volcanic eruptions that happened in this
period, with the 1900s being the only decade which featured a major volcanic
eruption in the Northern Hemisphere (Table 5) and which did not feature a poem that
considered the destructive nature of volcano-human interactions (again noting that
the 1940s and 1960s are not represented in this analysis). This would indicate that
even if poets were not writing about specific volcanic events, their attitudes might





have been affected by them (and the social and economic impacts that these large
volcanic events resulted in), leading to poems that more readily considered the
'Destructive' rather than 'Creative' nature of such interactions.
The 'Spiritual' category occurs across the considered time period, with a notable gap
in the first half of the twentieth century. This might in part be explained by the
tensions between poetry and religion that had manifested themselves throughout the
Victorian age (because of the constraints in the selection of the poetry discussed in
Section 2.2, the majority of the poems from this era came from either British or
American writers), with religion becoming increasingly personal and secular as
society became more democratic (Fraser, 1986). Similarly, the re-appearance of this
category in later poems corresponds to the emergence of a new Romantic
movement that began in the 1960s, in which people (and poets) began identifying
themselves as 'spiritual but not religious' (Thomas, 2006), which could explain the
re-emergence of 'Spirituality' in the poetry of this period.
**3.7 An emergent theme**
In considering the five categories identified in this analysis of the poetry, a clear
theme emerges with respect to RQ (what does poetry written about volcanoes reveal
about the relationship between humanity and volcanoes?): that there is a strong
sense of identity associated between humans and volcanoes.
An analysis of these poems has revealed that humans and volcanoes occupy the
same 'Landscape', and that whilst there is a certain reverence associated with their
'Destruction', there are also many positives associated with living alongside them.
Surprisingly however, these positives are mainly linked to notions of 'Identity' (e.g.
family histories and cultural heritages) rather than the physical benefits of 'Creation'
(e.g. fertile soils). These poems also highlight that this is a unidirectional relationship,
in part because of the distancing brought about by the associated 'Spirituality' or
otherness of the volcanoes, with humans ultimately of neither benefit nor concern to
the volcanoes that they write about.
This emergent theme is further supported by the quantitative analysis discussed in
Section 3.6, where over the time period considered there is a relative ubiquity of the
'Identity' to emerge from the poems. Similarly, despite the major cultural and societal
changes that have occurred over this time period, attitudes towards the human-
volcano interaction have not noticeably shifted in favour of one category or another;
the volcanoes have remained an ever present, suggesting that they are impervious
to these changes and further highlighting their unidirectional relationship with
humanity.





## 4. Conclusions

The purpose of this study was to investigate what poetry written about volcanoes reveals about the relationship between humanity and volcanoes. By conducting a qualitative content analysis of a selection of poems written from the 1800s to the present day, a series of themes emerged that characterised this relationship. The volcanoes as 'Landscape' and 'Identity' were seen as dominant, with consideration also given to the 'Destruction' and 'Spirituality' associated with these human-volcano interactions, and to a lesser extent their potential for 'Creation'. The main theme to emerge, both from these categories and the poems themselves, is that there is a strong sense of identity associated with volcanoes by humans (e.g. in terms of family histories, cultural heritage etc.), and that it is the humans (and not the volcanoes) that are affected by this relationship. A quantitative analysis of the frequency with which these categories occurred throughout the decades of this dataset supported the findings of this qualitative analysis, thereby further improving the validity and reliability of the main finding of this study, i.e. that the relationship between humanity and volcanoes is unidirectional and focused on identity.

The outcomes of this study support other research findings which have demonstrated that many communities are willing to accept the associated risks of living near a volcano, in order to experience the cultural and societal benefits (i.e. 'Identity') that they afford (see e.g. Schmincke, 2004;Kelman and Mather, 2008). Furthermore, by asking us to consider the emergent unidirectional nature of this relationship, this study also challenges us to re-consider the importance of humanity in our interactions with volcanoes. Unlike many other elements of our natural environment that have a strong sense of cultural and social identity attached to them (for example, glaciers, rivers, and rainforests), volcanoes are unlikely to be affected by anthropogenic climate change. Indeed, aside from humans, volcanism is itself a key driver in short-term climatic variations (Robock, 1991), with the net result a cooling at the Earth's surface due to the scattering of incoming solar radiation by secondary sulphate aerosols formed from volcanic eruptions (Cole-Dai, 2010).

The main limitation for this study is that only English-language poems were considered. This means that there is likely a bias towards certain attitudes or behaviours, especially those that were found to emerge from the 'Spirituality' category. Future research could, and should, include poetry written in multiple languages to account for this limitation, as doing so would reveal a broader understanding of how poets interpret human-volcano interactions, especially for communities from the Southern Hemisphere.

In addition to future studies considering a wider variety of languages and cultures, such work might also consider how different attitudes towards human-volcano interactions are captured by those poets who have physically encountered a volcano





versus those who are relying on second-hand testimony. Furthermore, by outlining how poems can be used as a form of data to provide further insight into how human-volcano interactions are perceived, this study suggests that a similar approach might also be adopted for other geoscientific events or phenomena.

We hope that this study has demonstrated that poetry is a powerful medium through which to consider perceptions of our natural environment, and that the results might better inform our communication with communities living nearby active volcanoes. It is in fact critical that local cultural and religious beliefs be taken into account when communicating volcanic hazard, as demonstrated by several social volcanology studies (Cashman and Cronin, 2008;Donovan, 2010;Lavigne et al., 2008;Paton et al., 2008). We welcome interested readers to get in contact with us for participation in the next stage of this research, in which we hope to broaden our analysis beyond the constraints of the English language.



**Data Availability**

The poems that were selected for the analysis, along with their coded categories, are available through Soldati and Illingworth (2019; https://doi.org/10.17605/OSF.IO/2D5K6).

**Author Contribution**

AS and SI worked together to conceive the design of this study, conduct the research and analysis, and write the paper.

**Competing interests**

Author SI is the chief executive editor of *Geoscience Communication*.

**Acknowledgements**

We would like to thank members of the Volcanology community who provided us with several poems to consider for our dataset. Furthermore, we would like to thank Alana Weir who provided useful comments to an earlier draft of this manuscript.





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





# Tables

Table 1: classification of the 41 poems according to their portrayal of the relationship between volcanoes and humanity as either 'Positive', 'Negative', 'Positive & Negative', 'Neutral' or 'Invalid'. Here the rows represent who the coding was performed by ('AS' - Arianna Soldati; 'SI' - Sam Illingworth; 'Both' - Arianna Soldati and Sam Illingworth).

| Coder | Positive | Negative | Positive & Negative | Neutral | Invalid |
|-------|----------|----------|---------------------|---------|---------|
| AS | 5 | 8 | 7 | 10 | 11 |
| SI | 13 | 16 | 0 | 9 | 3 |
| Both | 9 | 12 | 5 | 8 | 7 |

Table 2: the codes that emerged from an individual content analysis of the 34 poems, arranged according to the number of times they occurred ('AS' - Arianna Soldati; 'SI' - Sam Illingworth). *The number of occurrences is not limited to one per poem.

| Code | Coder | Description | Occurrences* |
|------|-------|-------------|--------------|
| Landscape | AS | Simple description of volcanoes as part of the environment | 30 |
| Human | AS | Referring to fellow human beings | 28 |
| Destruction | AS | Associated with destruction | 24 |
| Spiritual | AS | Evoking a spiritual connection or being | 23 |
| Destruction | SI | Something that brings / is associated with destruction | 15 |
| Creation | SI | Associated with new life and fostering life | 13 |
| Plants | AS | Reference to plants / flowers | 13 |





| | | | |
|---|---|---|---|
| **Spiritual** | **SI** | Associated with spiritual / religious feelings and/or presence | **12** |
| **History** | **AS** | Bringing a historical perspective | **12** |
| **Death** | **AS** | Associated with death | **12** |
| **Animals** | **AS** | Reference to animals | **11** |
| **Identity** | **SI** | Associated with a sense of identity for a person / people | **8** |
| **Fear** | **SI** | Something to be feared / afraid of | **8** |
| **Cohabiting** | **SI** | Living alongside the volcano (both positive and negative consequences) | **8** |
| **Natural Beauty** | **SI** | Described as something beautiful and/or awesome. | **7** |
| **Setting** | **SI** | The volcano is a neutral setting for the poem | **6** |
| **Universe** | **AS** | Reference to stars/planets | **5** |
| **Force / Power** | **AS** | Evoking the strength of the volcano | **5** |
| **Childhood** | **AS** | Reminiscing of childhood | **4** |
| **Peace** | **AS** | Relating to peace/absence of war | **3** |
| **Positive Metaphor** | **SI** | The volcano is used as a positive metaphor / simile | **2** |
| **Creation** | **AS** | Evoking the creation of land or life | **2** |
| **Rocks / Minerals** | **AS** | Reference to the rock/mineral element of landscape | **2** |



| Wish | AS | Expressing a desire, wish | 2 |
| Time | AS | Encompassing the passage of time | 2 |
| Quiet | AS | Related to a feeling of stillness, quiet | 2 |
| Pain | AS | Related to physical pain | 1 |
| Fear | AS | Related to a feeling of fear | 1 |

Table 3: the categories that emerged from the coding process, alongside their
corresponding codes, and arranged according to the number of times they occurred
('AS' - Arianna Soldati; 'SI' - Sam Illingworth).  *The number of occurrences is not
limited to one per poem. **This code originally had both positive and negative
connotations.

| Category | Corresponding Codes (Author) | Occurrences* |
|---|---|---|
| Landscape | Landscape (AS); Natural Beauty (SI); Universe (AS); Rocks/Minerals (AS); Force / Power (AS); Animals (AS); Plants (AS); Setting (SI) | 61 |
| Identity | History (AS); Heritage (AS); Humans (AS); Identity (SI); Childhood (AS); Positive Metaphor (SI); Time (AS) | 49 |
| Destruction | Death (AS); Pain (AS); Destruction (S); Destruction (AS); Fear (SI); Cohabiting** (SI); Fear (AS) | 36 |
| Spiritual | Spiritual (AS); Spiritual (SI) | 25 |
| Creation | Creation (SI); Creation (AS); Wish (AS); Cohabiting** (S); Peace (AS); Quiet (AS) | 8 |

Table 4. Poem connotation and categories by decade. In this table each decade was
considered to have an associated connotation if at least one poem written in this
decade was considered to be either 'Neutral', 'Positive', 'Negative', or 'Both'.
Similarly, a particular decade was assumed to be associated with an emergent
category if at least one of the poems that were written in that decade was
categorised as such.



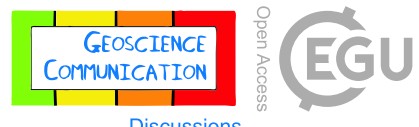

| Decade | Connotation | | | | Categories | | | | |
|---|---|---|---|---|---|---|---|---|---|
| | Neutral | Both | Positive | Negative | Landscape | Identity | Creation | Destruction | Identity |
| **1800** | x | | | | x | x | | x | |
| **1810** | | | | x | x | x | x | x | |
| **1820** | x | | | | x | x | | | x |
| **1830** | x | | | | X | x | | | x |
| **1840** | | x | | | x | x | x | x | x |
| **1850** | | | | x | x | x | | | x |
| **1860** | | | | x | x | | | x | |
| **1870** | | | | x | x | | | x | |
| **1880** | | x | | | x | | | x | x |
| **1890** | | | | x | x | x | | x | x |
| **1900** | | | x | | x | x | | | |
| **1910** | | | x | | x | x | | x | |
| **1920** | | x | | x | x | x | x | x | |
| **1930** | | | | x | x | x | | x | |
| **1940** | | | | | | | | | |
| **1950** | | x | | | x | x | | | |
| **1960** | | | | | | | | | |
| **1970** | x | | x | | x | x | x | | x |




| | | | | | | | | | |
|---|---|---|---|---|---|---|---|---|---|
| **1980** | | | x | x | x | x | x | x | x |
| **1990** | | | x | | x | x | x | x | x |
| **2000** | | | | x | x | x | | x | x |
| **2010** | x | | x | x | x | x | | | x |

Table 5. Major eruptions from the 1800s to the present day. These are defined as
those eruptions having a Volcanic Explosivity Index (VEI) greater than three and
which occurred in the Northern Hemisphere, where the majority of the authors of the
considered poems lived. Data taken from the Global Volcanism Program (Venzke,
7  2013).

| Volcano | Eruption Year |
|---|---|
| Tambora | 1815 |
| Krakatoa | 1883 |
| La Soufrière, Pelée, Santa Maria | 1902 |
| Novarupta | 1912 |
| Paricutin | 1943-54 |
| Vesuvius | 1944 |
| Surtsey | 1963 |
| St Helens | 1980 |
| Redoubt | 1989 |
| Pinatubo | 1991 |
| Eijafjallajokull | 2010 |

