# Peer review of "In my remembered country: what poetry tells"

_Geoscience Communication, 2019_

## Referee Comment (RC1) · David Pyle (Referee) · 3 Sep 2019

This paper explores the question of what poetry written about volcanoes reveals about the relationship between humans and volcanoes, using a small selection of English-language poems written since 1800. While the idea is certainly interesting, and the qualitative analysis does bring out some themes for discussion, my concern as a reader is that the analysis is obscured by the small number of poems under study, and the way they have been selected.

The analysis looks at 34 poems, written since 1800, predominantly by white male Anglophone poets. The time distribution is biased towards the present day. Of the twelve

19th century poets, two are women; and only 1 is a native of a volcanic land (Hallgrimson, Iceland). From the 20th century selection, five are women (one of whom is a volcano scientist); and six are natives of volcanic lands (Chile, Nicaragua/El Salvador, Hawaii).

It is not clear that the authors used a systematic approach to locating poems: was the 'manual internet search' simply on google? (and what were the search terms?); or did they use any of the databases of poems that might be accessible through library catalogues? Do the internal biases within the selections render invalid the idea that the poems can be used to 'tell us about changing perceptions of volcanoes'? What do we know, for example, about the first-hand experiences that the various writers had of volcanoes, or of volcanic activity? My instinct is that a more focussed analysis of a body of work that was better defined in terms of time and location, and considered critically in terms of the nature and experience of the author might provide additional insights into the research question.

Detailed points. 1. There is a body of relevant work which the authors don't cite but might consider.

Analysis of poetry from Montserrat: Donovan, A et al., 2011, Rationalising a volcanic crisis through literature: Montserratian verse and the descriptive reconstruction of an island, Journal Of Volcanology And Geothermal Research, 2011 Jun 15(3-4), pp.87-101. Skinner, J, 2011, A Distinctive Disaster Literature: Montserrat Island Poetry under Pressure, in Islanded Identities, Constructions of Postcolonial Cultural Insularity, Cross/Cultures, Volume: 139, https://doi.org/10.1163/9789401206938_004

Victorian Disaster Poetry: Altick, RD (1960) Four Victorian Poets and an Exploding Island, Victorian Studies, Vol. 3, No. 3 (Mar., 1960), pp. 249-260 (12 pages), https://www.jstor.org/stable/3825498

2. (page 3) Introduction: there are over 1400 'active volcanoes' and 50 – 60 in eruption in any given year (Smithsonian GVP catalogue; https://volcano.si.edu/); the areal footprint of an eruption doesn't scale in a simple way with VEI; and the reporting of past eruptions was much more about where they occurred, than their size: every burst of activity at Vesuvius was reported from the eighteenth century and on; meanwhile, the effects of the 'great eruption of Tambora' was barely known about until decades later.

3. Page 3 line 42 – see also: Hamilton, J., 2012, Volcano: nature and culture. Reaktion Books, London, 2012; Alexander, D., 2016, The portrayal of disaster in Western fine art, Environmental Hazards, Volume 15, Pages 209-226, https://doi.org/10.1080/17477891.2016.1173007

4. Page 4 line 4 – 'vog' is a localised or tropospheric phenomenon; most of the sunset colours are a consequence of stratospheric particulates.

5. Page 5, line 17 – it would be of considerable value to also have an appendix to the paper that lists the poems in this study.

6. Results and discussion: it would be worthwhile analysing where and by whom the poems were written?

7. Page 11, line 27 – it's not quite true that La Soufriere 'dominates' the physical landscape of St Vincent; it can hardly be seen from most of the island. I'd agree that it dominates the metaphorical landscape.

8. Page 15, line 13 – it is interesting that you chose to focus on the Christian element of the tale of St Telemachus' death; a more obvious volcano link comes from the opening lines 'HAD the fierce ashes of some fiery peak / Been hurl'd so high they ranged about the globe?' which refer to the eruption of Krakatoa.

9. Page 17, line 8; page 19 line 11 – in 1816 Byron would not have known that the dismal weather had a volcanic cause; this didn't become known until decades later, and the eruption of Krakatoa. Here the 'mountain-torch' is a reference to the way that Byron imagined a volcano might light up the gloom.

10. Table 5 – this is a curious list. Tambora and Krakatoa are both in the southern

hemisphere (but had global effects); and the list of eruptions is (surely) far from complete – even at a threshold of VEI 5 (e.g. Cosiguina, Nicaragua, 1835; El Chichon, Mexico, 1982), What about Hekla? And other major eruptions of Vesuvius?

11. It might be appropriate to follow standard procedure for citing poems, by referring to the line numbers in the excerpts?

––––––––––––––––––––––––––––––––

---

## Referee Comment (RC2) · Christos S. Zerefos (Referee) · 4 Oct 2019

Human volcano relationships have been discussed in a number of papers in the past. They include interaction with colors and at sunsets of large volcanic eruptions. Poetry has not been used so far and the paper presents a beautiful selection of poetry analyzed in an innovative way. A quantitative analysis is quite convincing and useful in an emerging new era that brings closer science and art.

---

## Referee Comment (RC3) · Anonymous Referee #3 · 10 Oct 2019

This study presents a method to evaluate the changing perceptions of volcanoes and present its results when applied to a pool of 34 poems written between 1800-pst. This is an interesting study, which however requires in my view several significant improvements.

**General comments**

**1. The title does not reflect well the study: it is too general for the collection of poems considered**
As the authors indicate, they had to impose several constraints (time period, language,

length, preferred writing-style, etc), resulting in a collection of only 34 poems, which is a very limited pool of poems for such an ambitious title.

In addition to the importance of the language/culture restriction, limitation underlined in the conclusion of the manuscript, the time period considered (1800 - present), selected because it coincides with the emergence and development of volcanalogy, might also have a strong impact on the final results. In Europe/US (where the majority of the artists come from), this period coincides with many scientific discoveries and social changes (related to the industrialisation) compared with previous centuries, which modified in particular spiritual/religious believes and human-Nature relationship, two themes which are considered in this study. Hence, the results regarding the temporal evolution of the various categories, would have likely been different if the time period considered would have included centuries prior to the $19^{th}$. For these reasons, the title should be more specific (or the pool of poems studied extended).

**2. The manuscript (particularly introduction and methodology) would benefit from various clarifications**:

- **what is the exact object of the study ? Is the study about the perception of active volcanoes or of all volcanoes ?**
  The introduction develops several arguments related to active volcanoes, however some of the poems considered concerns non-erupting volcanoes (eg as mentioned in the section about "Landscape")

- **Please clarify throughout the manuscript that the results are about the poets' perception of the interaction between Men-volcano (although it can be generalised to a certain extent)**
  For example the way the RQ is formulated does not give an active role to the poets' subjectivity ("RQ: what does poetry written about volcanoes reveal about the relationship between humanity and volcanoes?", **p5, l30-31**), contrarily to the

aims of the study given just before ("understand the way in which poets have interpreted the relationship between humans and volcanoes" (**p5, l26-27**))

**Specific comments**

**p3, l13-19**
Please clarify the main message of this paragraph. Is it to justify the choice of "significant eruption" (Table 5) ? The eruptions impacting the most human societies are not necessarily those with bigger VEI, but that population can be affected significantly by small eruptions and associated hazards. With the development of medias, transportation, etc, information about an eruption -even small- can also reach much more people.

**Please in the introduction give explicit examples of various ways in which volcano and the link humanity-volcano can be perceived ?**
This could be done re-writing the section **p4, l1-9** ("We can conceivably use art pieces to guide us in our understanding of how volcano-human relations have changed through time. [...]"), highlighting what the artworks demonstrates of the volcano-artist interaction, rather than eg describing the painting and the physical phenomenon.

**p8, l40-42 "However, as outlined by Morse et al. (2002), our methodological coherence, sampling strategy, and saturation of emergent codes ensures the reliability, and thus repeatability, of our approach in this qualitative analysis."**
Although I agree this approach would give similar results if the same two authors were two repeat the experiment, how can you ensure the results would be similar if the study would be carried out by two (or more) people, with potentially different backgrounds, etc and hence likely to interpret the poems differently ?

**p11, l10 "the poets perceived very clearly their [the volcanoes'] relationship with**

**humanity"; p21, l32-33 "with humans ultimately of neither benefit nor concern to the volcanoes that they write about":**
In several instances throughout the manuscript, such as in thes examples, the authors personify the volcanoes. Please could you modify the manuscript such that it is clear in your analysis that the only subject(s) is/are the poet/humans and that all human emotions assigned to volcanoes are only perceptions. In the first example above, the sentence could be modified to eg "the poets perceived very clearly how humanity relates to volcanoes" (ie volcanoes are objects)

**p11, l13-14 "poetry acts to position volcanoes as an awesome part of our shared landscape, perhaps explaining in part why humans were first drawn to them before they became valued for more tangible goods":**
I suggest you suppress this assumption. Overall, population settlements are unlikely to have been drawn primarily by the beauty of an area. The people attracted by the spectacle given by active volcanoes are usually not the people who live nearby, and potentially risk their life and their families', livelihood, properties, etc.

**p20, l21-22 "many communities are willing to accept the associated risks of living near a volcano, in order to experience the cultural and societal benefits"**
Could you precise if that is for active volcanoes ? During the 1995-pst eruption of Soufriere Hills, Montserrat, 75% of the population left the island which would contradict the statement above. Those would stayed, did not for the cultural and societal benefits, but mostly for either economical reasons, or because they could not imagine loosing their roots.

**p22, l24-25: "the emergent unidirectional nature of this relationship, this study also challenges us to re-consider the importance of humanity in our interactions with volcanoes.Unlike many other elements of our natural environment that have a strong sense of cultural and social identity attached to them (for example,**

**glaciers, rivers, and rainforests)"**
Please include in your **introduction** a short paragraph (perhaps at the very beginning
of the paper) mentioning studies (possibly as yours) about interactions between other
natural elements and humans

**p22, l28 "by anthropogenic climate change"**
Anthropogenic climate change is not the only way humans affect the environment.

**p22, l28-31 "Indeed, aside from humans, volcanism is itself a key driver in short-
term climatic variations (Robock, 1991), with the net result a cooling at the
Earth's surface due to the scattering of incoming solar radiation by secondary
sulphate aerosols formed from volcanic eruptions (ColeDai, 2010).":**
Please suppress, this is outside of the topic.

**p23, l7-9 "We hope that this study has demonstrated that poetry is a powerful
medium through which to consider perceptions of our natural environment, and
that the results might better inform our communication with communities living
nearby active volcanoes.**
Please suppress "We hope [...] that', there is no need to demonstrate that, however
could you indicate more practically how the results could be used to work with commu-
nities ?

**Technical corrections**

**p3,l11 "Certainly, they do not leave us indifferent"**
Please suppress this comment: that is the authors' opinion, however it can surely not
be applied to everybody, even those leaving near an active volcano.

**p3, l33 "yet while these works have great merit"**
Please suppress, it sounds condescending

**p3, l35 "volcanic activity clashes with human life"**
There is probably a less informal term than "clash" eg impact, affect, etc

**p8, l39 " described here represents a somewhat subjective approach"**
Please suppress "somewhat"

**p18, l3 "respectful of the human space"**
No need to change your interpretation, but isn't it just a way to show that humanity can't be at peace, even when everything else is 'purified' by fire ?

**Figures 1 and 4:**
It would improve their readability (in particular the years) and help the comparisons between categories, if you would merge the various subfigures in one figure. For example you can use a color per category. Please also indicate what decades were not represented.

**p21, l6-8: This might in part be explained by the tensions between poetry and religion that had manifested themselves throughout the Victorian age**
Also, society and arts were impacted by WWI and WWII.

**p21, l23 "that there is a strong sense of identity associated between humans and volcanoes":**
Could you rephrase more clearly please ?

---

## Referee Comment (RC4) · Christos S. Zerefos (Referee) · 10 Oct 2019

On the general comments of Referee #3:

Comment 1: The number of 44 poems considered limited by Referee #3, I don't think that can limit the importance of that interesting note. I have to agree that maybe the period referred (19th century to present) could be added in the title.

Comment 2: Bullet 1- My impression when I read the manuscript was that it referred to large volcanic eruptions as listed in Table 5 of the manuscript.

---

## Author Comment (AC1) · 12 Nov 2019

We thank the reviewer for this positive feedback!
* * *

---

## Author Comment (AC2) · 12 Nov 2019

David Pyle (Referee)

This paper explores the question of what poetry written about volcanoes reveals about the relationship between humans and volcanoes, using a small selection of English language poems written since 1800. While the idea is certainly interesting, and the qualitative analysis does bring out some themes for discussion, my concern as a reader is that the analysis is obscured by the small number of poems under study, and the way they have been selected. The analysis looks at 34 poems, written since 1800, predominantly by white male Anglophone poets. The time distribution is biased towards

the present day. Of the twelve 19th century poets, two are women; and only 1 is a native of a volcanic land (Hallgrimson, Iceland). From the 20th century selection, five are women (one of whom is a volcano scientist); and six are natives of volcanic lands (Chile, Nicaragua/El Salvador, Hawaii).

We recognize these limitations. The choice of Anglophone poets is intentional (p.5 l.39) and its limitations are acknowledged (p.22 l.33). This of course affects the volcanic land native/non-native demographics. The same goes for the historical period, as justified starting on p.5 l.42. The gender bias is indeed present, and we now explicitly acknowledge it (p. 6 l.29).

It is not clear that the authors used a systematic approach to locating poems: was the 'manual internet search' simply on google? (and what were the search terms?); or did they use any of the databases of poems that might be accessible through library catalogues?

We actually first used three databases: poetry foundation, poetry society, and poetry archive – then we conducted more specific Internet searches to fill decade gaps. We have now clearly detailed this methodology in the manuscript (p.6 l.7). We thank the reviewer for highlighting the need for a more thorough description of our search methodology.

Do the internal biases within the selections render invalid the idea that the poems can be used to 'tell us about changing perceptions of volcanoes'? What do we know, for example, about the first-hand experiences that the various writers had of volcanoes, or of volcanic activity? My instinct is that a more focussed analysis of a body of work that was better defined in terms of time and location, and considered critically in terms of the nature and experience of the author might provide additional insights into the research question.

We believe they do not – we have clearly stated the limitations of our study, and our conclusions rest on them. However, we are very interested in expanding the study in

future work. For example, we had already mentioned our intention to work on the first-vs. second-hand experience of volcanoes (p.23 l.1).

Detailed points.

1. There is a body of relevant work which the authors don't cite but might consider. Analysis of poetry from Montserrat: Donovan, A et al., 2011, Rationalising a volcanic crisis through literature: Montserratian verse and the descriptive reconstruction of an island, Journal Of Volcanology And Geothermal Research, 2011 Jun 15(3-4), pp.87- 101. Skinner, J, 2011, A Distinctive Disaster Literature: Montserrat Island Poetry under Pressure, in Islanded Identities, Constructions of Postcolonial Cultural Insularity, Cross/Cultures, Volume: 139, https://doi.org/10.1163/9789401206938_004 Victorian Disaster Poetry: Altick, RD (1960) Four Victorian Poets and an Exploding Island, Victorian Studies, Vol. 3, No. 3 (Mar., 1960), pp. 249-260 (12 pages), https://www.jstor.org/stable/3825498 2.

We thank the reviewer for bringing these works to our attention. We have now cited them in the Introduction.

2. Introduction: there are over 1400 'active volcanoes' and 50 – 60 in eruption in any given year (Smithsonian GVP catalogue; https://volcano.si.edu/); the areal footprint of an eruption doesn't scale in a simple way with VEI; and the reporting of past eruptions was much more about where they occurred, than their size: every burst of activity at Vesuvius was reported from the eighteenth century and on; meanwhile, the effects of the 'great eruption of Tambora' was barely known about until decades later.

This is a good point, and we have now incorporated it on p.3 l.7

3. Page 3 line 42 – see also: Hamilton, J., 2012, Volcano: nature and culture. Reaktion Books, London, 2012; Alexander, D., 2016, The portrayal of disaster in Western fine art, Environmental Hazards, Volume 15, Pages 209-226, https://doi.org/10.1080/17477891.2016.1173007 4.

We thank the reviewer for suggesting these works – we have referenced them.

4. Page 4 line 4 – 'vog' is a localised or tropospheric phenomenon; most of the sunset colours are a consequence of stratospheric particulates.

Indeed. We thank the reviewer for catching this mistake, which we have now rectified.

5. Page 5, line 17 – it would be of considerable value to also have an appendix to the paper that lists the poems in this study.

We created a DOI that contains this information. It is open access, and is now more clearly signposted in the manuscript.

6. Results and discussion: it would be worthwhile analysing where and by whom the poems were written?

Yes, that will be the subject of a follow up study. We have already started gathering the relevant information, and we are looking forward to developing this research further once this first study is published.

7. Page 11, line 27 – it's not quite true that La Soufriere 'dominates' the physical landscape of St Vincent; it can hardly be seen from most of the island. I'd agree that it dominates the metaphorical landscape.

Thank you for pointing out the need for clarification, we have now specified the metaphorical connotation of that sentence.

8. Page 15, line 13 – it is interesting that you chose to focus on the Christian element of the tale of St Telemachus' death; a more obvious volcano link comes from the opening lines 'HAD the fierce ashes of some fiery peak / Been hurl'd so high they ranged about the globe?' which refer to the eruption of Krakatoa.

We coded everything in the poem, as can be seen in the supplementary material. In this section we are just providing examples of the different emerging categories. For "spiritual", St. Telemachus contains a representative one, which we chose to highlight.

9. Page 17, line 8; page 19 line 11 – in 1816 Byron would not have known that the dismal weather had a volcanic cause; this didn't become known until decades later, and the eruption of Krakatoa. Here the 'mountain-torch' is a reference to the way that Byron imagined a volcano might light up the gloom.

We agree that thus point needed clarification, which we have now provided. We found it interesting that, although Byron didn't know the cause, he felt this way about the consequences.

10. Table 5 – this is a curious list. Tambora and Krakatoa are both in the southern hemisphere (but had global effects); and the list of eruptions is (surely) far from complete – even at a threshold of VEI 5 (e.g. Cosiguina, Nicaragua, 1835; El Chichon, Mexico, 1982), What about Hekla? And other major eruptions of Vesuvius?

Yes, this is not a complete list of all volcanic eruptions that occurred over the considered time span, not even with a VEI 5 threshold. We compiled this list keeping in mind the impact (including the mediatic resonance) that these eruptions would have had. For example, as the reviewer notes, two eruptions occurred in the southern hemisphere, but had global effects, and in fact they were written about by poets. We have added the El Chichon and Hekla eruptions to the list. We have also rephrased the Table caption, which was misleading.

11. It might be appropriate to follow standard procedure for citing poems, by referring to the line numbers in the excerpts?

Given that this study is submitted for consideration to a Geoscience journal, and that all cited excerpts are provided in-text, we have not made this addition.

---

## Author Comment (AC3) · 12 Nov 2019

General comments 1. The title does not reflect well the study: it is too general for the collection of poems considered. As the authors indicate, they had to impose several constraints (time period, language, paper length, preferred writing-style, etc), resulting in a collection of only 34 poems, which is a very limited pool of poems for such an ambitious title. In addition to the importance of the language/culture restriction, limitation underlined in the conclusion of the manuscript, the time period considered (1800 - present), selected because it coincides with the emergence and development of vol-

canalogy, might also have a strong impact on the final results. In Europe/US (where the majority of the artists come from), this period coincides with many scientific discoveries and social changes (related to the industrialisation) compared with previous centuries, which modified in particular spiritual/religious believes and human-Nature relationship, two themes which are considered in this study. Hence, the results regarding the temporal evolution of the various categories, would have likely been different if the time period considered would have included centuries prior to the 19th. For these reasons, the title should be more specific (or the pool of poems studied extended).

We have expanded the title to read: "In my remembered country: what poetry tells us about the changing perceptions of volcanoes between the XIX and XXI centuries". This will help the potential reader quickly gauge their interest in the study. Further details can be found in the abstract.

2. The manuscript (particularly introduction and methodology) would benefit from various clarifications: • what is the exact object of the study ? Is the study about the perception of active volcanoes or of all volcanoes ? The introduction develops several arguments related to active volcanoes, however some of the poems considered concerns non-erupting volcanoes (eg as mentioned in the section about "Landscape")

It is the perception of all volcanoes – nowhere do we state that that we focus on either erupting or active volcanoes. We have however specified so in Section 2.1.

• Please clarify throughout the manuscript that the results are about the poets' perception of the interaction between Men-volcano (although it can be generalised to a certain extent) For example the way the RQ is formulated does not give an active role to the poets' subjectivity ("RQ: what does poetry written about volcanoes reveal about the relationship between humanity and volcanoes?", p5, l30-31), contrarily to the aims of the study given just before ("understand the way in which poets have interpreted the relationship between humans and volcanoes" (p5, l26-27))

In order to reconcile this apparent discrepancy, in Section 2.1 we have now specified

that we consider poets to be representatives of humankind.

Specific comments p3, l13-19 Please clarify the main message of this paragraph. Is it to justify the choice of "significant eruption" (Table 5)? The eruptions impacting the most human societies are not necessarily those with bigger VEI, but that population can be affected significantly by small eruptions and associated hazards. With the development of medias, transportation, etc, information about an eruption -even small- can also reach much more people. Please in the introduction give explicit examples of various ways in which volcano and the link humanity-volcano can be perceived ? This could be done re-writing the section

This paragraph is indeed meant to lean into the choice of eruptions presented in Table 5. However, we are still in the introduction section here. We are not claiming that eruptions with bigger VEI impact human societies the most (or the most human societies), just that their aerial impact footprint are larger. We have expanded this paragraph to comment on the impact of small eruptions, and to highlight the role of communication means (p.3 l.17).

p4, l1-9 ("We can conceivably use art pieces to guide us in our understanding of how volcano-human relations have changed through time. [...]"), highlighting what the artworks demonstrates of the volcano-artist interaction, rather than eg describing the painting and the physical phenomenon.

We have added a couple of references to clarify this point.

p8, l40-42 "However, as outlined by Morse et al. (2002), our methodological coherence, sampling strategy, and saturation of emergent codes ensures the reliability, and thus repeatability, of our approach in this qualitative analysis." Although I agree this approach would give similar results if the same two authors were two repeat the experiment, how can you ensure the results would be similar if the study would be carried out by two (or more) people, with potentially different backgrounds, etc and hence likely to interpret the poems differently ?

Thank you for raising this important point. With qualitative research, different strategies in comparison to quantitative research must be adopted in order to ensure the credibility of the study findings. As you point out, it may be that other researchers arrive at slightly different conclusions, however this is the nature of all research. What we have done in this study is to maintain a 'decision trail', so that the reader (and any future researchers) can follow our analysis and the clear and transparent decisions that we have provided. However, we take your point that 'repeatability' might not be the most appropriate word here, and so the text has been changed to account for this and now reads: "... our methodological coherence, sampling strategy, and saturation of emergent codes ensures the reliability and trustworthiness of our approach in this qualitative analysis."

p11, l10 "the poets perceived very clearly their [the volcanoes'] relationship with humanity"; p21, l32-33 "with humans ultimately of neither benefit nor concern to the volcanoes that they write about": In several instances throughout the manuscript, such as in thes examples, the authors personify the volcanoes. Please could you modify the manuscript such that it is clear in your analysis that the only subject(s) is/are the poet/humans and that all human emotions assigned to volcanoes are only perceptions. In the first example above, the sentence could be modified to eg "the poets perceived very clearly how humanity relates to volcanoes" (ie volcanoes are objects)

We have rephrased as suggested to make it clearer.

p11, l13-14 "poetry acts to position volcanoes as an awesome part of our shared landscape, perhaps explaining in part why humans were first drawn to them before they became valued for more tangible goods": I suggest you suppress this assumption. Overall, population settlements are unlikely to have been drawn primarily by the beauty of an area. The people attracted by the spectacle given by active volcanoes are usually not the people who live nearby, and potentially risk their life and their families', livelihood, properties, etc.

We chose the term "awesome" carefully, because it designates something (in our case volcanoes) that inspires awe and is overpowering – not necessarily something that is beautiful.

p20, l21-22 "many communities are willing to accept the associated risks of living near a volcano, in order to experience the cultural and societal benefits" Could you precise if that is for active volcanoes ? During the 1995-pst eruption of Soufriere Hills, Montserrat, 75% of the population left the island which would contradict the statement above. Those would stayed, did not for the cultural and societal benefits, but mostly for either economical reasons, or because they could not imagine loosing their roots.

It applies to both active (now précised) and inactive volcanoes. Economic reasons are also societal benefits, and "could not imagine loosing their roots" is a cultural benefit, as we have explored in our analysis.

p22, l24-25: "the emergent unidirectional nature of this relationship, this study also challenges us to re-consider the importance of humanity in our interactions with volcanoes.Unlike many other elements of our natural environment that have a strong sense of cultural and social identity attached to them (for example, glaciers, rivers, and rainforests)" Please include in your introduction a short paragraph (perhaps at the very beginning of the paper) mentioning studies (possibly as yours) about interactions between other natural elements and humans

It would not be correct to include this in the introduction; this aspect emerged through the study, and we did not start with a pre-conception that that might happen. As such, we believe that this belongs where it is, in the conclusions. We have added references to three studies about interactions between other natural elements and humans.

p22, l28 "by anthropogenic climate change" Anthropogenic climate change is not the only way humans affect the environment.

That was only an example, and we have now specified that.

p22, l28-31 "Indeed, aside from humans, volcanism is itself a key driver in shortterm climatic variations (Robock, 1991), with the net result a cooling at the Earth's surface due to the scattering of incoming solar radiation by secondary sulphate aerosols formed from volcanic eruptions (ColeDai, 2010).": Please suppress, this is outside of the topic.

Point taken, we have removed that sentence.

p23, l7-9 "We hope that this study has demonstrated that poetry is a powerful medium through which to consider perceptions of our natural environment, and that the results might better inform our communication with communities living nearby active volcanoes. Please suppress "We hope [...] that', there is no need to demonstrate that, however could you indicate more practically how the results could be used to work with communities?

We removed "We hope". Additionally, we provided a practical example of how the results could be used to work with communities (p.23 l.10).

Technical corrections

p3,l11 "Certainly, they do not leave us indifferent" Please suppress this comment: that is the authors' opinion, however it can surely not be applied to everybody, even those leaving near an active volcano.

It can be vastly applied indeed. We added a reference that supports our statement.

p3, l33 "yet while these works have great merit" Please suppress, it sounds condescending

We did not mean to be condescending. We have suppressed that in order to avoid any misunderstandings.

p3, l35 "volcanic activity clashes with human life" There is probably a less informal term than "clash" eg impact, affect, etc

We changed "clashes" to "conflicts".

p8, l39 " described here represents a somewhat subjective approach" Please suppress "somewhat"

We cannot suppress "somewhat", as that would introduce an error: there are also objective elements in the analyses – e.g. the identification of very concrete themes (plants, animals...)

p18, l3 "respectful of the human space" No need to change your interpretation, but isn't it just a way to show that humanity can't be at peace, even when everything else is 'purified' by fire ?

We respectfully disagree with this interpretation. Moreover, there is no fire in volcanoes.

Figures 1 and 4: It would improve their readability (in particular the years) and help the comparisons between categories, if you would merge the various subfigures in one figure. For example you can use a color per category. Please also indicate what decades were not represented.

After preparing a new figure following the reviewer's suggestion, we feel that readability is much better in the current version.

p21, l6-8: This might in part be explained by the tensions between poetry and religion that had manifested themselves throughout the Victorian age Also, society and arts were impacted by WWI and WWII.

We thank the reviewer for the insight, but we feel that this takes the explanation too far. We were just offering one example. We do however welcome further research of course.

p21, l23 "that there is a strong sense of identity associated between humans and volcanoes": Could you rephrase more clearly please ?

The meaning has been extensively explained before in Section 3.2, which is entirely dedicated to this theme and contains several examples.

---

## Author Comment (AC4) · 12 Nov 2019

Comment 1: The number of 44 poems considered limited by Referee #3, I don't think that can limit the importance of that interesting note. I have to agree that maybe the period referred (19th century to present) could be added in the title.

Thank you, we agree. We have indeed added the period referred to the title.

Comment 2: Bullet 1- My impression when I read the manuscript was that it referred to large volcanic eruptions as listed in Table 5 of the manuscript.

Correct, and we have now made that more explicit.

---

## Referee Report (RR1)

[referee-annotated manuscript omitted]

---

## Author Response (AR2)

**Anonymous reviewer:**

My main comment remains, that is the limited pool of poems used to question the "relationship between humanity and volcanoes" (since XIXe century). In order to address this comment, I suggest that

1. either you include more poems in your study
2. or that you repeat in your "Conclusions" section -as you already did for the language- the limitations in terms of representativity related to the limited number of poems/poets considered and the poets' gender (as highlighted during the review process), as well as the assumption that the poets reflects their social environment

*Option 1 cannot be pursued, because as we explained in the methodology section our selection of poems represents all poems that could be found that match our criteria. We therefore revised our conclusions session to include the study limitations and assumptions*

**David Pyle:**

The authors have not fully taken on board the depth or breadth of the points raised by two of the reviewers, and this version of this paper doesn't fully explain to the reader the preliminary nature of the analysis and its potential limitations. Being upfront about these limitations will only the strengthen the paper - as it will help to point the way to future work.

*We agree that being upfront about the limitations of the study will only the strengthen the paper, and we have now been even more explicit and detailed than before – please see answers to detailed comments below. However, the analysis presented in this paper is not preliminary: much future work can stem from it, but it will have a different focus (e.g. consider different languages, the poets' scientific knowledge, etc.), as opposed to expanding on the same research question.*

*Additionally, we have reframed the RQ as: "what does poetry written about volcanoes reveal about the relationship between volcanoes and the societies and times represented by poets who wrote about them?"*

Two key points:
- the sample size of poems analysed really is still too small for any meaningful 'quantitative' analysis [it really is not quantative]; and many 'decades' are exemplified by one poem, or fewer.

*As explained in the methodology section, the sample size of poems analyzed reflects all of the poems that are available and that match our criteria. We did indeed perform a quantitative (as opposed to qualitative) analyses of the poems in part of the work. However, the reviewer's point made us understand the need to explicitly mention that although it is quantitative, our analysis is not statistically significant, and that for this to be the case a larger dataset would indeed be needed. This information is included on page 18, lines 37-38: "Due to the limited sample size (n=34), this quantitative analysis cannot be considered statistically significant."*

- the absence of any discussion of the way the cultural, social, political.. backdrop to the poems has also changed through time is a severe limitation; has writing about nature also changed over the same timescale? And what might be the effect (on the time progression) of having 19thC authors who are privileged northern-European men (like Lord Byron), and late 20thC authors who include women who live in volcanic countries?

*We agree that this is indeed a limitation, and we have now clearly mentioned this fact on page 23, lines 6-12:*

*"It is also worth noting that throughout the 220 years considered in this study, the cultural, social, and political backdrop of the poems has changed significantly. For example, colonialism has the potential of having significantly affected the representation of cultural elements in poetry. Whereas a detailed analyses of how these changes affected the content and tone of the poems is beyond the scope of this work, future studies focussing on this aspect would certainly be valuable."*

*Analyzing the evolution in time of backdrop to the poems' detail would require too much background, i.e. European history over two centuries, which is beyond the scope of this paper. This would however be very a very interesting topic for a dissertation or future study. The fact that gender representation is not equitable was already mentioned (page 6, lines 32-34). We have now reinforced this limitation on page 23 lines 3-5, where we have also added that it changes throughout our sample with time:*

*"Additionally, the majority of poems considered were authored by male poets, due to the scarcity of poems about volcanoes written by female poets, especially for the 19th Century."*

*Furthermore, our dataset does not provide any indication that the evolving gender balance of the poets may affect either the connotations (Fig. 1) or themes (Fig. 2) of the poems. Finally, we already mentioned that analyses of the direct experience of volcanoes will be covered in further studies on page 23, lines 14-17:*

*"In addition to future studies considering a wider variety of languages and cultures, such work might also consider how different attitudes towards human-volcano interactions are captured by those poets who have physically encountered a volcano versus those who are relying on second-hand testimony.".*

**Editor**

Their main points can be summarised as follows:
1. There are not enough sample poems in the analysis
2. The implications from the analysis are too large

One cannot extrapolate from 34 poems over 200 years to the perceptions of all humanity. Just because the poets are "part" of humanity does not mean that they "represent" humanity. You write in the article that the poets are "representatives of the society and cultures they live in". All that remains is that you frame the analysis and discussion accordingly. You need to remove any statements that indicate that these poems represent humanity as a whole, and tone down the implications of the study.

*We have reframed the study around the feelings of the societies (and times) the poets represent. The rephrased research question now reads: "what does poetry written about volcanoes reveal about the relationship between volcanoes and the societies and times represented by poets who wrote about them?"*

My other worry also stems from Prof. Pyle's comments, that there is an absence of "any discussion of the way the cultural, social, political.. backdrop to the poems has also changed through time is a severe limitation". This issue has also been referred to by referee #3. I have knowledge about how Hawaiian culture considers the volcanoes and thereby their goddess Pele. There are several cultures and power dynamics influencing what a single poet would write about when it comes to the volcanoes and the goddess Pele. Only an analysis of many poems, from many different authors, would be able to illustrate how humanity in Hawaii perceives volcanoes. You don't need to go into a deep analysis of these issues as they will be numerous the world over. However, you need to be aware of them and mention them. You also need to change your language so that you are aligned with these issues. Again, it's impossible to extrapolate from one poem mentioning Pele to how all "humanity" in Hawaii perceives the volcanoes.
*We have now addressed that on Page 23, lines 6-12.*

In addition (and with the Hawaiian example in mind), I believe it is a stretch to make the conclusion that "the relationship between humanity and volcanoes is unidirectional and focused on identity", "with humans ultimately of neither benefit nor concern to the volcanoes that they write about". In traditional Hawaiian culture, Pele (the volcano), gives so much to the people, and she is recognised in traditional song and dance. This does not strike me as unidirectional. The poems may indicate a unidirectional relationship, but hopefully this exemplifies that that does not reflect the perception of all of humanity in Hawaii. I am not expecting you to go into a deep analysis of the cultural power dynamics here. However, you have to be aware that volcanoes have huge cultural significance in some indigenous cultures. And as soon as this occurs, then it is almost impossible to avoid a discussion on (or at least mention of) colonialism.

*We now mention this on page 2-221, lines 43-2.*

In short, you need to emphasise that the poems are a product of the society and culture the poets are a part of. You therefore need to reframe your research question and implications accordingly

*This has now been done, please see answer to RQ reframing suggestion above.*

One final point from myself is that you state that "the results might better inform our communication with communities living nearby active volcanoes". Without concrete ideas about how this might happen, then I would ask you to remove this from the text. With only 34 poems in your sample, extrapolating the results to general communication strategies could actually be damaging, as the situation in Hawaii hopefully illustrates. But on the (very!) positive side, your methodology could be an integral part in a deeper analysis with a local or regional community to develop communication strategies.

*We simply meant that taking into account the local communities' views of the volcanoes, as captured by poetry, is an important point to consider. We understand the Editor's concern about the original wording, and we have rephrased as suggested.*

**Major changes**

By addressing all of the concerns of the reviewers, the main change in this version of the manuscript is that we have rephrased the research question as:

"what does poetry written about volcanoes reveal about the relationship between volcanoes and the societies and times represented by the poets who wrote about them?"

This better reflects the limitations of the study, and fully takes on board the reviewers' and editor's comments. We have consequently reframed the study's implications.

[revised manuscript text omitted]

---

## Author Response (AR3)

I am in general happy with the responses and alterations the authors have made in this round.
I believe that much of mine and the reviewers comments stem from the previous overreaching
research question. The authors have indeed altered the research question itself, but not followed
through with the rest of the article, as I clearly articulated previously.
The word "humanity" and it's relationship to the poems is still mentioned 11 times throughout
the text. This must be amended before I can accept for publication.

*That has now been removed from anywhere in the manuscript.*

[revised manuscript text omitted]